# SIMPLE AND EFFECTIVE VAE TRAINING WITH CALIBRATED DECODERS

## ABSTRACT

Variational autoencoders (VAEs) provide an effective and simple method for modeling complex distributions. However, training VAEs often requires considerable hyperparameter tuning to determine the optimal amount of information retained by the latent variable. We study the impact of calibrated decoders, which learn the uncertainty of the decoding distribution and can determine this amount of information automatically, on the VAE performance. While many methods for learning calibrated decoders have been proposed, many of the recent papers that employ VAEs rely on heuristic hyperparameters and ad-hoc modifications instead. We perform the first comprehensive comparative analysis of calibrated decoder and provide recommendations for simple and effective VAE training. Our analysis covers a range of datasets and several single-image and sequential VAE models. We further propose a simple but novel modification to the commonly used Gaussian decoder, which computes the prediction variance analytically. We observe empirically that using heuristic modifications is not necessary with our method.

## 1 INTRODUCTION

Deep density models based on the variational autoencoder (VAE) (Kingma & Welling, 2014; Rezende et al., 2014) have found ubiquitous use in probabilistic modeling and representation learning as they are both conceptually simple and are able to scale to very complex distributions and large datasets. These VAE techniques are used for tasks such as future frame prediction (Castrejon et al., 2019), image segmentation (Kohl et al., 2018), generating speech (Chung et al., 2015) and music (Dhariwal et al., 2020), as well as model-based reinforcement learning (Hafner et al., 2019a). However, in practice, many of these approaches require careful manual tuning of the balance between two terms that correspond to distortion and rate from information theory (Alemi et al., 2017). This balance trades off fidelity of reconstruction and quality of samples from the model: a model with low rate would not contain enough information to reconstruct the data, while allowing the model to have high rate might lead to unrealistic samples from the prior as the KL-divergence constraint becomes weaker (Alemi et al., 2017; Higgins et al., 2017). While a proper variational lower bound does not expose any free parameters to control this tradeoff, many prior works heuristically introduce a weight on the prior KL-divergence term, often denoted $\beta$. Usually, $\beta$ needs to be tuned for every dataset and model variant as a hyperparameter, which slows down development and can lead to poor performance as finding the optimal value is often prohibitively computationally expensive. Moreover, using $\beta \neq 1$ precludes the appealing interpretation of the VAE objective as a bound on the data likelihood, and is undesirable for applications like density modeling.

While many architectures for calibrating decoders have been proposed in the literature (Kingma & Welling, 2014; Kingma et al., 2016; Dai & Wipf, 2019), more applied work typically employs VAEs with uncalibrated decoding distributions, such as Gaussian distributions without a learned variance, where the decoder only outputs the mean parameter (Castrejon et al., 2019; Denton & Fergus, 2018; Lee et al., 2019; Babaeizadeh et al., 2018; Lee et al., 2018; Hafner et al., 2019b; Pong et al., 2019; Zhu et al., 2017; Pavlakos et al., 2019), or uses other ad-hoc modifications to the objective (Sohn et al., 2015; Henaff et al., 2019). Indeed, it is well known that attempting to learn the variance in a Gaussian decoder may lead to numerical instability (Rezende & Viola, 2018; Dai & Wipf, 2019), and naïve approaches often lead to poor results. As a result, it remains unclear whether practical empirical performance of VAEs actually benefits from calibrated decoders or not.

To rectify this, our first contribution is a comparative analysis of various calibrated decoder architectures and practical recommendations for simple and effective VAE training. We find that, while naïve calibrated decoders often lead to worse results, a careful choice of the decoder distribution can work very well, and removes the need to tune the additional parameter $\beta$. Indeed, we note that the entropy of the decoding distribution controls the mutual information $I(x; z)$. Calibrated decoders allow the model to control $I(x; z)$ automatically, instead of relying on manual tuning. Our second contribution is a simple but novel technique for optimizing the decoder variance analytically, without requiring the decoder network to produce it as an additional output. We call the resulting approach to learning the Gaussian variance the $\sigma$-VAE. In our experiments, the $\sigma$-VAE outperforms the alternative of learning the variance through gradient descent, while being simpler to implement and extend. We validate our results on several VAE and sequence VAE models and a range of image and video datasets.

## 2 RELATED WORK

Prior work on variational autoencoders has studied a number of different decoder parameterizations. Kingma & Welling (2014); Rezende et al. (2014) use the Bernoulli distribution for the binary MNIST data and Kingma & Welling (2014) use Gaussian distributions with learned variance parameter for grayscale images. However, modeling images with continuous distributions is prone to instability as the variance can converge to zero (Rezende & Viola, 2018; Mattei & Frellsen, 2018; Dai & Wipf, 2019). Some work has attempted to rectify this problem by using dequantization (Gregor et al., 2016), which is theoretically appealing as it is tightly related to the log-likelihood of the original discrete data (Theis et al., 2016), optimizing the variance in a two-stage procedure (Arvanitidis et al., 2017), or training a post-hoc prior (Ghosh et al., 2019). Takahashi et al. (2018); Barron (2019) proposed more expressive distributions. Additionally, different choices for representing such variance exist, including diagonal covariance (Kingma & Welling, 2014; Sønderby et al., 2016; Rolfe, 2016), or a single shared parameter (Kingma et al., 2016; Dai & Wipf, 2019; Edwards & Storkey, 2016; Rezende & Viola, 2018). We analyze these and notice that learning a single variance parameter shared across images leads to stable training and good performance, without the use of dequantization or even clipping the variance, although these techniques can be used with our decoders; and further improve the estimation of this variance with an analytic solution.

Early work on discrete VAE decoders for color images modeled them with the Bernoulli distribution, treating the color intensities as probabilities (Gregor et al., 2015). Further work has explored various parameterizations based on discretized continuous distributions, such as discretized logistic (Kingma et al., 2016). More recent work has improved expressivity of the decoder with a mixture of discretized logistics (Chen et al., 2016; Maaløe et al., 2019). However, these models also employ powerful autoregressive decoders (Chen et al., 2016; Gulrajani et al., 2016; Maaløe et al., 2019), and the latent variables in these models may not represent all of the significant factors of variation in the data, as some factors can instead be modeled internally by the autoregressive decoder (Alemi et al., 2017).[1]

While many calibrated decoders have been proposed, outside the core generative modeling community uncalibrated decoders are ubiquitous. They are used in work on video prediction (Denton & Fergus, 2018; Castrejon et al., 2019; Lee et al., 2018; Babaeizadeh et al., 2018), image segmentation (Kohl et al., 2018), image-to-image translation (Zhu et al., 2017), 3D human pose (Pavlakos et al., 2019), as well as model-based reinforcement learning (Henaff et al., 2019; Hafner et al., 2019b;a), and representation learning (Lee et al., 2019; Watter et al., 2015; Pong et al., 2019). Most of these works utilize the heuristic hyperparameter $\beta$ instead, which is undesirable both as the resulting objective is no longer a bound on the likelihood, and as $\beta$ usually requires extensive tuning. In this work, we analyze the common pitfalls of using calibrated decoders that may have prevented the practitioners from using them, propose a simple and effective analytic way of learning such calibrated distribution, and provide a comprehensive experimental evaluation of different decoding distributions.

Alternative discussions of the hyperparameter $\beta$ are presented by Zhao et al. (2017); Higgins et al. (2017); Alemi et al. (2017); Achille & Soatto (2018), who show that it controls the amount of information in the latent variable, $I(x; z)$. Peng et al. (2018); Rezende & Viola (2018) further discuss constrained optimization objectives for VAEs, which also yield a similar hyperparameter. Here, we focus on $\beta$-VAEs with Gaussian decoders with constant variance, as commonly used in recent work, and show that the hyperparameter $\beta$ can be incorporated in the decoding likelihood for these models.

---

[1] BIVA (Maaløe et al., 2019) uses the Mixture of Logistics decoder proposed in (Salimans et al., 2017) that produces the channels for each pixel autoregressively, see also App D.

# 3 ANALYSING DECODING DISTRIBUTIONS

The generative model of a VAE (Kingma & Welling, 2014; Rezende et al., 2014) with parameters $\theta$ is specified with a prior distribution over the latent variable $p_\theta(z)$, commonly unit Gaussian, and a decoding distribution $p_\theta(x|z)$, which for color images is commonly a conditional Gaussian parameterized with a neural network. We would like to fit this generative model to a given dataset by maximizing the evidence lower bound (ELBO (Neal & Hinton, 1998; Jordan et al., 1999; Kingma & Welling, 2014; Rezende et al., 2014)), which uses an approximate posterior distribution $q_\phi(z|x)$, also commonly a conditional Gaussian specified with a neural network. In this work, we focus on the form of the decoding distribution $p_\theta(x|z)$. To achieve the best results, we want a decoding distribution that represents the required probability $p(x|z)$ accurately In this section, we will review and analyze various choices of decoding distributions that enable better decoder calibration, including expressive decoding distributions that can represent both the prediction of the image and the uncertainty about such prediction, or even multimodal predictions.

## 3.1 GAUSSIAN DECODERS

We first analyse the commonly used Gaussian decoders. We note that the commonly used MSE reconstruction loss between the reconstruction $\hat{x}$ and ground truth data $x$ is equivalent to the negative log-likelihood objective with a Gaussian decoding distribution with constant variance:

$$-\ln p(x|z) = \frac{1}{2}||\hat{x} - x||^2 + D\ln\sqrt{2\pi} = \frac{1}{2}||\hat{x} - x||^2 + c = \frac{D}{2}\text{MSE}(\hat{x}, x) + c,$$

where $p(x|z) \sim \mathcal{N}(\hat{x}, I)$, the prediction $\hat{x}$ is produced with a neural network $\hat{x} = \mu_\theta(z)$, and $D$ is the dimensionality of $x$.

This demonstrates a drawback of methods that rely simply on the MSE loss (Castrejon et al., 2019; Denton & Fergus, 2018; Lee et al., 2019; Hafner et al., 2019b; Pong et al., 2019; Zhu et al., 2017; Henaff et al., 2019), as it is equivalent to assuming a particular, constant variance of the Gaussian decoding distribution. By learning this variance, we can achieve much better performance due to better calibration of the decoder. There are several ways in which we can specify this variance. An expressive way to specify the variance is to specify a diagonal covariance matrix for the image, with one value per pixel (Kingma & Welling, 2014; Sønderby et al., 2016; Rolfe, 2016). This can be done, for example, by letting a neural network $\sigma_\theta$ output the diagonal entries of the covariance matrix given a latent sample $z$:

$$p_\theta(x|z) \sim \mathcal{N}\left(\mu_\theta(z), \sigma_\theta(z)^2\right). \tag{1}$$

This parameterization of the decoding distribution outputs one variance value per each pixel and channel. While powerful, we observe in Section 5.3 that this approach attains suboptimal performance, and is moreover prone to numerical instability. Instead, we will find experimentally that a simpler parameterization, in which the covariance matrix is specified with a single shared (Kingma et al., 2016; Dai & Wipf, 2019; Edwards & Storkey, 2016; Rezende & Viola, 2018) parameter $\sigma$ as $\Sigma = \sigma I$ often works better in practice:

$$p_{\theta,\sigma}(x|z) \sim \mathcal{N}\left(\mu_\theta(z), \sigma^2 I\right). \tag{2}$$

The parameter $\sigma$ can be optimized together with parameters of the neural network $\theta$ with gradient descent. Of particular interest is the interpretation of this parameter. Writing out the expression for the decoding likelihood, we obtain

$$-\ln p(x|z) = \frac{1}{2\sigma^2}||\hat{x}-x||^2 + D\ln\sigma\sqrt{2\pi} = \frac{1}{2\sigma^2}||\hat{x}-x||^2 + D\ln\sigma + c = D\ln\sigma + \frac{D}{2\sigma^2}\text{MSE}(\hat{x}, x) + c.$$

The full objective of the resulting Gaussian $\sigma$-VAE is:

$$\mathcal{L}_{\theta,\phi,\sigma} = D\ln\sigma + \frac{D}{2\sigma^2}MSE(\hat{x}, x) + D_{KL}(q(z|x)||p(z)). \tag{3}$$

Note that $\sigma$ may be viewed as a weighting parameter between the MSE reconstruction term and the KL-divergence term in the objective. Moreover, this objective explicitly specifies how to select the optimal variance: the variance should be selected to minimize the (weighted) MSE loss while also minimizing the logarithm of the variance.

**Decoder Calibration** It is important that the decoder distribution be calibrated in the statistical sense, that is, the predicted probabilities should correspond to the frequencies of seeing a particular value of $x$ given that prediction (DeGroot & Fienberg, 1983; Dawid, 1982). The calibration of a neural network can be usually improved by estimating the uncertainty of that prediction (Guo et al., 2017), such as the variance of a Gaussian (Kendall & Gal, 2017). Since the naive MSE loss assumes a constant variance, it does not effectively represent the uncertainty of the prediction, and is often poorly calibrated. Instead, learning the variance as in Eq. 3 leads to better uncertainty estimation and better calibration. In Sec 5.1, we show that learning a good estimate of this uncertainty is crucial for the quality of the VAE generations.

**Connection to $\beta$-VAE.** The $\beta$-VAE objective (Higgins et al., 2017) for a Gaussian decoder with unit variance is:

$$\mathcal{L}^{\beta} = \frac{D}{2} MSE(\hat{x}, x) + \beta D_{KL}(q(z|x)||p(z)). \tag{4}$$

We see that it can be interpreted as a particular case of the objective (3), where the variance is constant and the term $D \ln \sigma$ can be ignored during optimization. The $\beta$-VAE objective is then equivalent to a $\sigma$-VAE with a constant variance $\sigma = \sqrt{\beta/2}$ (for a particular learning rate setting). In recent work (Zhu et al., 2017; Denton & Fergus, 2018; Lee et al., 2019), $\beta$-VAE models are often used in this exact regime. By tuning the $\beta$ term, practitioners are able to tune the variance of the decoder, manually producing a more calibrated decoder. However, by re-interpreting the $\beta$-VAE objective as a special case of the VAE and introducing the missing $D \ln \sigma$ term, we can both obtain a valid evidence lower bound, and remove the need to manually select $\beta$. Instead, the variance $\sigma$ can instead simply be learned end-to-end, reducing the need for hyperparameter tuning.

An alternative discussion of this connection in the context of linear VAEs is also presented by Lucas et al. (2019). While the $\beta$ term is not necessary for good performance if the decoder is calibrated, it can still be employed if desired, such as when the aim is to attain better disentanglement (Higgins et al., 2017) or a particular rate-distortion tradeoff (Alemi et al., 2017). However, we found that with calibrated decoders, the best sample quality is obtained when $\beta = 1$.

**Loss implementation details.** For the correct evidence lower bound computation, it is necessary to add the values of the MSE loss and the KL divergence across the dimensions. We observe that common implementations of these losses (Denton & Fergus, 2018; Abadi et al., 2016; Paszke et al., 2019) use averaging instead, which will lead to poor results if the number of image dimensions is significantly different from the number of the latent dimensions. While this can be conveniently ignored in the $\beta$-VAE regime, where the balance term is tuned manually anyway, for the $\sigma$-VAE it is essential to compute the objective value correctly.

**Variance implementation details.** Since the variance is non-negative, we parameterize it logarithmically as $\sigma^2 = e^{2\lambda}$, where $\lambda$ is the logarithm of the standard deviation. For some models, such as per-pixel variance decoders, we observed that it is necessary to restrict the variance range for numerical stability. We do so by using the soft clipping operations proposed by Chua et al. (2018):

$$\lambda := \lambda_{\max} - \text{softplus}(\lambda_{\max} - \lambda); \qquad \lambda := \lambda_{\min} + \text{softplus}(\lambda - \lambda_{\min}).$$

We observe that setting $\lambda_{\min} = -6$ to lower bound the standard deviation to be at least half of the distance between allowed color values works well in practice. We also observe that this clipping is unnecessary when learning a shared $\sigma$ value.

### 3.2 DISCRETE DECODERS

It is possible to use discrete decoding distributions to generate images, as color values are commonly restricted to a fixed set of integer pixel intensities (e.g. 0..255). Indeed, for discrete color values, discrete distributions are arguably more appropriate. In the most general case, a discrete decoding distribution factorized per each pixel and channel would be specified by a probability mass vector $\hat{x}$ with 256 entries, one per each possible intensity value, similarly to a per-pixel classifier of the intensity value. We can implement it with a soft-max layer, yielding the following log-likelihood loss (sometimes called the cross-entropy loss) for a true pixel with intensity $i$:

$$-\ln p(x|z) = -\ln \frac{\exp(\hat{x}_i)}{\sum_j \exp(\hat{x}_j)},$$

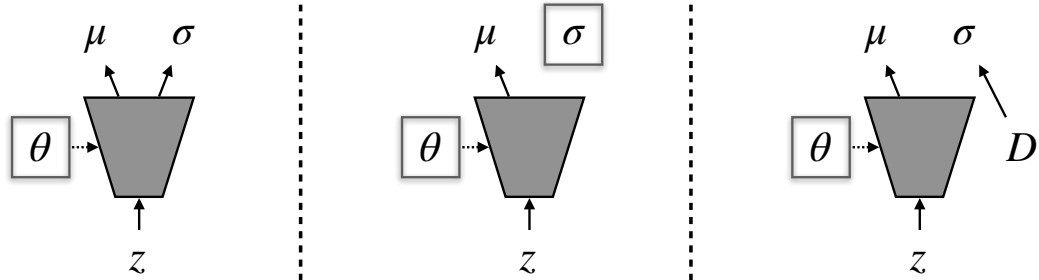

Figure 1: Different types of calibrated decoders for Gaussian VAE, model parameters are denoted with enclosing squares. Left: both the mean $\mu$ and the variance $\sigma$ are output by a neural network with parameters $\theta$. Center: $\sigma$-VAE with shared variance, the mean is output by a neural network with parameters $\theta$, but the variance it iself a global parameter. Right: the proposed optimal $\sigma$-VAE, the mean is output by a neural network with parameters $\theta$, and the variance is computed analytically from the training data $D$.

We will evaluate these and further choices of discrete decoders, described in Appendix D. We recommend choosing the decoder distribution that best suits the structure of the data, such as discrete decoders for discrete data and continuous decoders for continuous data.

## 4   OPTIMAL VARIANCE ESTIMATION FOR CALIBRATED GAUSSIAN DECODERS

In this section, we propose a simple but novel analytic way of obtaining a calibrated decoder for continuous distributions that further improves performance. The Gaussian decoders with learned variance described in Section 3.1 are calibrated and work better than naïve unit variance decoders. However, for $\sigma$-VAE optimized with gradient descent or Adam (Kingma & Ba, 2015), we observe that careful learning rate tuning can yield significantly better performance, which is in line with prior work that reported poor performance of gradient descent for optimizing Gaussian distributions (Amari, 1998; Peters & Schaal, 2008). A smaller learning rate often produces better performance, but slows down the training, as the likelihood values $p(x|z)$ will be very suboptimal in the beginning. Instead, here we propose an analytic solution for the value of $\sigma$, which computes it analytically and does not require gradient descent.

The maximum likelihood estimate of the variance given a known mean is the average squared distance from the mean:

$$\sigma^* = \arg\max_{\sigma} \mathcal{N}(x|\mu, \sigma^2 I) = \text{MSE}(x, \mu), \tag{5}$$

where $\text{MSE}(x, \mu) = \frac{1}{D}\sum_i (x_i - \mu_i)^2$. Eq. 5 can be easily shown using manual differentiation, and is a generalization of the fact that the MLE estimate of the variance is the sample variance.

The optimal variance for the decoder distribution under the maximum likelihood criterion is then simply the average MSE loss over the data and the encoder distribution. We leverage this to create an optimal analytic solution for the variance. In the batch setting, the optimal variance would be simply the MSE loss, and can be updated after every gradient update for the other parameters of the decoder. In the mini-batch setting, we use a batchwise estimate of the variance computed for the current minibatch. We analyze these approximations in Appendix C. At test time, a running average of the variance over the training data is used. This method, which we call *optimal $\sigma$-VAE*, allows us to learn very efficiently as we use the optimal variance estimate at every training step. It is also easier to implement, as no separate optimizer for the variance parameter is needed. If the variance is not needed at test time, it can also be simply discarded after training.

**Per-image optimal $\sigma$-VAE.**   Optimal $\sigma$-VAE uses a single variance value shared across all data points. However, the optimal $\sigma$-VAE also allows more powerful variance estimates, such as learning a variance value per each pixel, or even a variance value per each image, the difference in implementation simply being the dimensions across which the averaging in Equation 5 operates. This approach can be interpreted as variational variance prediction in the framework of Stirn & Knowles (2020).

## 5   EXPERIMENTAL RESULTS

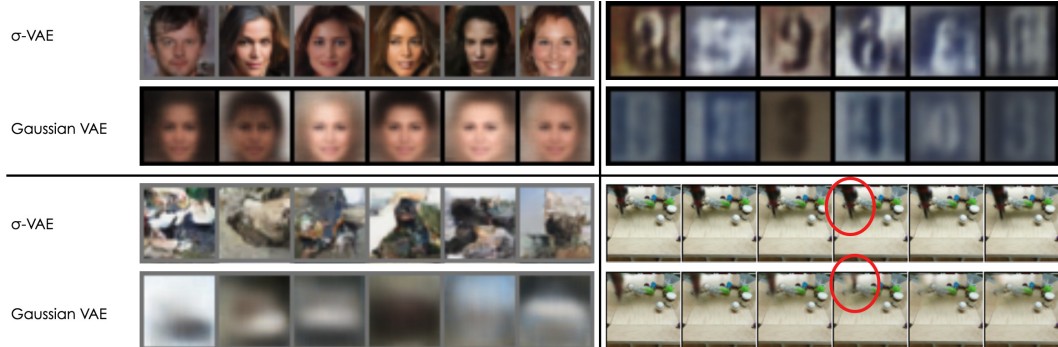

Figure 2: Images or videos (bottom right) sampled from the proposed optimal $\sigma$-VAE and a unit variance Gaussian VAE models. The Gaussian VAE does not have a means to control the expressivity of the latent variable and produces suboptimal, blurry samples. The $\sigma$-VAE controls the expressivity by learning a calibrated decoder, and produces higher quality sequences on all datasets.

We now provide an empirical analysis of different decoding distributions, and validate the benefits of our $\sigma$-VAE approach. We use a small convolutional VAE model on SVHN (Netzer et al., 2011), a larger hierarchical HVAE model (Maaløe et al., 2019) on the CelebA (Liu et al., 2015) and CIFAR (Krizhevsky et al., 2009) datasets, and a sequence VAE model called SVG (Denton & Fergus, 2018) on the BAIR Pushing dataset (Finn & Levine, 2017). We evaluate the ELBO values as well as visual quality measured by the Fréchet Inception Distance (FID, Heusel et al. (2017)). Images are $28 \times 28$ for SVHN and $32 \times 32$ for CelebA and CIFAR, while video experiments were performed on $64 \times 64$ frames

Table 1: Analysis of learned variance on SVHN. The parameter $\beta$ is tuned manually in $\beta$-VAE and learned in $\sigma$-VAE. $\sigma$-VAE achieves better performance, while the value of $\beta$ (implicitly defined via the decoder variance) automatically converges close the value found by manual tuning.

|  | $\beta$ | $-\log p \downarrow$ | FID $\downarrow$ |
|---|---|---|---|
| $\beta$-VAE | 0.001 | $< 21.43$ | 44.54 |
| $\beta$-VAE | 0.01 | $< -3186$ | 27.93 |
| $\beta$-VAE | 0.1 | $< -1223$ | 28.3 |
| $\beta$-VAE | 1 | $< 1381$ | 70.39 |
| $\beta$-VAE | 10 | $< 4056$ | 219.3 |
| $\sigma$-VAE | 0.006 | $< -\mathbf{3333}$ | **22.25** |

following Denton & Fergus (2018). We do not use KL annealing as it did not improve the results in our experiments. Further experimental details are in App. B.

### 5.1 DO CALIBRATED DECODERS BALANCE THE VAE OBJECTIVE WITHOUT TUNING $\beta$?

As detailed in Section 3.1, a $\beta$-VAE with a unit variance Gaussian decoder commonly used in prior work is equivalent to a $\sigma$-VAE with constant, manually tuned variance. There is a simple relationship between beta and the variance: $\sigma = \sqrt{\beta/2}$. To compare the variance that the $\sigma$-VAE learns to the manually tuned variance in the case of the $\beta$-VAE, we compare the ELBO values and the corresponding values of $\beta$ in Table 1. We find that learning the variance produces similar values of $\beta$ to the manually tuned values in the $\beta$-VAE case, indicating that the $\sigma$-VAE is able to learn the balance between the two objective terms in a single training run, without hyperparameter tuning. Moreover, the $\sigma$-VAE outperforms the best $\beta$-VAE run. This is because end-to-end learning produces better estimates of

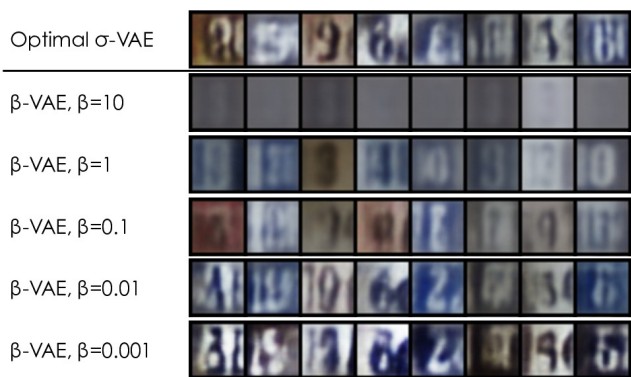

Figure 3: Analysis of learned variance on SVHN. The parameter $\beta$ is tuned manually in $\beta$-VAE and learned in $\sigma$-VAE. Higher values of $\beta$ cause the images to lose detail, while lower values of $\beta$ might make samples unrealistic. The proposed optimal $\sigma$-VAE is able to learn the balance end-to-end, here converging to an equivalent of $\beta$-VAE with $\beta = 0.006$.

the variance than is possible with manual search, improving the likelihood (as measured by the lower bound) and the visual quality. Figure 3 shows the qualitative results from this experiment.

We further validate our results on both single-image and sequential VAE models on a range of datasets in Table 2 and Figure 2. Single-sample ELBO values are reported, and ELBO values on discretized data are reported for discrete distributions. We see that learning a shared variance in a Gaussian decoders (shared $\sigma$-VAE) outperforms the naïve unit variance decoder (Gaussian VAE) as well as tuning the $\beta$ constant for the Gaussian VAE manually. We also see that calibrated discrete decoders, such as full categorical distribution or mixture of discretized logistics, perform better than the naïve Gaussian VAE. Using Bernoulli distribution by treating the color intensities as probabilities (Gregor et al., 2015; Watter et al., 2015) performs poorly. Our results further improve upon the sequence VAE method of Denton & Fergus (2018), which uses a unit variance Gaussian with the $\beta$-VAE objective.

## 5.2 HOW DOES LEARNING CALIBRATED DECODERS IMPACT THE LATENT VARIABLE INFORMATION CONTENT?

We saw above that calibrated decoders result in higher log-likelihood bounds. Are calibrated decoders also beneficial for representation learning? We evaluate the mutual information $I_e(x; z)$ between the data $p_d(x)$ and encoder samples $q(z|x)$, as well as the mismatch between the prior $p(z)$ and the marginal encoder distribution $m(z) = E_{p_d(x)}q(z|x)$, measured by the marginal KL $D_{KL}(m(z)||p(z))$. These terms are related to the rate term of the VAE objective as follows (Alemi et al., 2017):

$$E_{p_d(x)}\left[D_{KL}(q(z|x)||p(z))\right] = E_{p_d(x)}\left[D_{KL}(q(z|x)||m(z))\right] + D_{KL}(m(z)||p(z))$$
$$= I_e(x; z) + D_{KL}(m(z)||p(z)). \tag{6}$$

That is, the rate term decomposes into the true mutual information and the marginal KL term. We want to learn expressive latent variables with high mutual information. However, doing so by tuning the $\beta$ value relaxes the constraint that the encoder and the prior distributions match, and leads to degraded quality of samples from the prior, which creates a trade-off between expressive representations and ability to generate good samples. To compare the $\beta$-VAE and $\sigma$-VAE in terms of these quantities, we estimate the marginal KL term via Monte Carlo sampling, as proposed by Rosca et al. (2018), and plot the results in Figure 4. As expected, we see that lower $\beta$ values lead to higher mutual information. However, after a certain point, lower values of $\beta$ also cause a significant mismatch between the marginal and the prior distributions. By calculating the "effective" $\beta$ for the $\sigma$-VAE, as per Section 4, we can see that the $\sigma$-VAE captures an inflection point in the $D_{KL}(m(z)||p(z))$ term, learning a representation with the highest possible MI, but without degrading sample quality. This explains the high visual quality of the optimal $\sigma$-VAE samples: since the marginal and the prior distributions match, the samples from

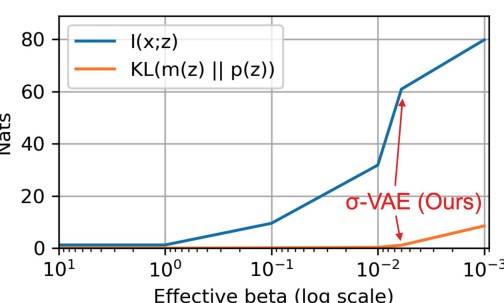

Figure 4: Comparison of $\beta$-VAE and $\sigma$-VAE on SVHN in terms of mutual information $I_e(x; z)$ and marginal KL divergence $KL(m(z)||p(z))$ (see Sec. 5.2). $I_e(x; z)$ increases with lower $\beta$, yielding expressive representations and better reconstruction. However, after a certain point, lowering $\beta$ leads to a rapid increase in the marginal KL, yielding poor samples from the prior. The $\sigma$-VAE is able to automatically find the inflection point after which the marginal KL begins to increase, capturing as much information as possible while still producing good samples.

the prior look similar to reconstructions, while for a $\beta$-VAE with low $\beta$, the samples from the prior are poor. We see that, in contrast to the $\beta$-VAE, where the mutual information is controlled by a hyperparameter, the $\sigma$-VAE can adjust the appropriate amount of information automatically and is able to find the setting that produces both informative latents and high quality samples.

An alternative discussion of tuning $\beta$ is presented by Alemi et al. (2017), who show that $\beta$ controls the rate-distortion trade-off. Here, we show that the crucial trade-off also controlled by $\beta$ is the

Table 2: Generative modeling performance of the proposed $\sigma$-VAE on different models and datasets. For SVG, we compare with the original method (Denton & Fergus, 2018), which uses $\beta$-VAE. We see that uncalibrated decoders such as mean-only Gaussian perform poorly. $\beta$-VAE allows to calibrate the decoder but needs careful hyperparameter tuning. Calibrated decoders such as categorical or $\sigma$-VAE perform best. [1] Gregor et al. (2015), [2] Takahashi et al. (2018), [3] Higgins et al. (2017).

| | CelebA HVAE | | SVHN VAE | | CIFAR HVAE | | BAIR SVG | |
| --- | --- | --- | --- | --- | --- | --- | --- | --- |
| | $-\log p \downarrow$ | FID $\downarrow$ | $-\log p \downarrow$ | FID $\downarrow$ | $-\log p \downarrow$ | FID $\downarrow$ | $-\log p \downarrow$ | FID $\downarrow$ |
| Bernoulli VAE [1] | | 177.6 | | 43.26 | | 284.5 | | 122.6 |
| Categorical VAE | < **6359** | 71.5 | < 9179 | 46.13 | < **7179** | **101.7** | N/A | N/A |
| Bitwise-categorical VAE | < 9067 | 66.61 | < 10800 | 33.84 | < 9390 | **91.2** | < 48744 | 46.13 |
| Logistic mixture VAE | < 7932 | 65.3 | < **9085** | 43.19 | < 8443 | 143.1 | < **40616** | 42.94 |
| Gaussian VAE | < 7173 | 186.5 | < 2184 | 112.5 | < 7186 | 293.7 | < −10379 | 35.64 |
| Per-pixel $\sigma$-VAE | < −7814 | 159.3 | < 2184 | 114.7 | < −7222 | 131 | < −14051 | 41.98 |
| Student-t VAE [2] | < −8401 | 71.06 | < **−3659** | 70.4 | < **−7419** | 123.6 | - | - |
| $\beta$-VAE [3] | < −2713 | **61.6** | < −3186 | 27.93 | < −331 | **103** | < −13472 | 34.64 |
| Shared $\sigma$-VAE | < −6374 | **60.7** | < −3349 | **22.25** | < −5435 | 116.1 | < −13974 | 34.24 |
| Optimal $\sigma$-VAE | < **−8446** | **60.3** | < −3333 | 27.25 | < −5677 | **101.4** | < **−14173** | 34.13 |
| Opt. per-image $\sigma$-VAE | | 66.01 | | 26.28 | | **104.0** | | **33.21** |

trade-off between two components of the rate itself, which control expressivity of representations and the match between the variational and the prior distributions, respectively.

### 5.3 WHAT ARE THE COMMON CHALLENGES IN LEARNING THE VARIANCE THAT PREVENT PRACTITIONERS FROM USING IT, AND HOW TO RECTIFY THEM?

If learning the decoder variance improves generation, why are learned variances not used more often? In this section, we discuss how the naïve approach to learning variances, where the decoder outputs a variance for each pixel along with the mean, leads to poor results. First, we find that this method often diverges very quickly due to numerical instability, as the network is able to predict certain pixels with very high certainty, leading to degenerate variances. In contrast, learning a shared variance is always numerically stable in our experiments. We can rectify this numerical instability by bounding the output variance (Section 3.1). However, even with bounded variance, we observe that learning per-pixel variances leads to poor results in Table 2. While the per-pixel variance achieves a good ELBO value, it produces very poor samples, as measured by FID and visual inspection.

We see that the specific form of learned variance: a shared variance, a per-image variance, or a per-pixel variance, can lead to very different performance in practice. We hypothesize the per-pixel decoder performs poorly as it incentivizes the model to focus on particular pixels that can be predicted well, instead of focusing equally on all parts of the image. This is consistent with prior work on denoising diffusion models which noted that likelihood-based models place too much focus on imperceptible details, which leads to deteriorated results (Ho et al., 2020). The shared and per-image variance models mitigate this issue at the cost of introducing more bias, and work better in practice.

### 5.4 CAN AN ANALYTIC SOLUTION FOR OPTIMAL VARIANCE FURTHER IMPROVE LEARNING?

We evaluate the *optimal $\sigma$-VAE* which uses an analytic solution for the variance (Section 4). Table 2 shows that it achieves superior results in terms of log-likelihood. We also note that the optimal $\sigma$-VAE converges to a good variance estimate instantaneously, which speeds up learning (highlighted in Figure 9 in the Appendix). In addition, we evaluate the per-image optimal $\sigma$-VAE, in which a single variance is computed per image. This model achieves significantly higher visual quality. While producing this per-image variance with a neural network would require additional architecture tuning, optimal $\sigma$-VAE is extremely simple to implement (it can be implemented simply as changing the axes of summation), not requiring any new tunable parameters.

## 6 CONCLUSION

We presented a simple and effective method for learning calibrated decoders, as well as an evaluation of different decoding distributions with several VAE and sequential VAE models. The proposed

method outperforms methods that use naïve unit variance Gaussian decoders and tune a heuristic weight $\beta$ on the KL-divergence loss, as commonly done in prior work. Moreover, it does not use the heuristic weight $\beta$, making it easier to train than this prior work. We expect that the simple techniques for learning calibrated decoders can allow practitioners to speed up the development cycle, obtain better results, and reduce the need for manual hyperparameter tuning.

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

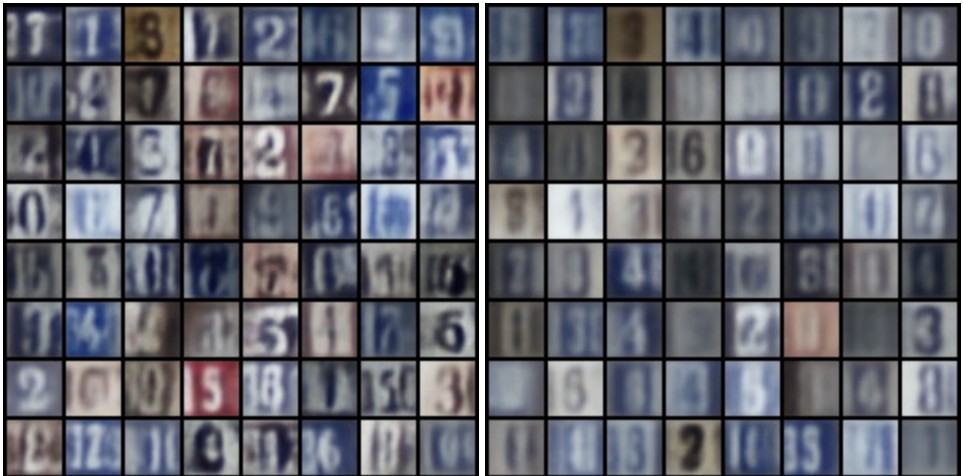

Figure 5: Samples from the $\sigma$-VAE (left) and the Gaussian VAE (right) on the SVHN dataset. The Gaussian VAE produces blurry results with muted colors, while the $\sigma$-VAE is able to produce accurate images of digits.

Lucas Theis, Aäron van den Oord, and Matthias Bethge. A note on the evaluation of generative models. ICLR, 2016.

Manuel Watter, Jost Springenberg, Joschka Boedecker, and Martin Riedmiller. Embed to control: A locally linear latent dynamics model for control from raw images. In Advances in neural information processing systems, pp. 2746–2754, 2015.

Shengjia Zhao, Jiaming Song, and Stefano Ermon. Infovae: Information maximizing variational autoencoders. arXiv preprint arXiv:1706.02262, 2017.

Jun-Yan Zhu, Richard Zhang, Deepak Pathak, Trevor Darrell, Alexei A Efros, Oliver Wang, and Eli Shechtman. Toward multimodal image-to-image translation. In Advances in neural information processing systems, pp. 465–476, 2017.

## A    ADDITIONAL EXPERIMENTAL RESULTS

In this section, we provide more qualitative results in Figures 7, 6, 8, 5 as well as a graph showing the convergence properties of the variance for different models in Fig. 9. In order to validate our method with a different architecture, we also report performance of different decoders with a small 5-layer convolutional architecture on the CelebA and CIFAR dataset in Table 3. We see that the ordering of the methods is consistent with this smaller architecture.

## B    EXPERIMENTAL DETAILS

For the small convolutional network test on SVHN, the encoder has 3 convolutional layers followed by a fully connected layer, while the decoder has a fully connected layer followed by 3 convolutional layers. The $\beta$ was tuned from 100 to 0.0001 for $\beta$-VAE. The number of channels in the convolutional layers starts with 32 and increases 2 times in every layer. The dimension of the latent variable is 20. Adam (Kingma & Ba, 2015) with learning rate of 1e-3 is used for optimization. Batch size of 128 was used and all models were trained for 10 epochs. We additionally evaluate this small convolutional network on CelebA, CIFAR, and Frey Face[2] datasets in Table 3. Unit Gaussian prior and Gaussian posteriors with diagonal covariance were used. For the larger hierarchical VAE, we used the official pytorch implementation of (Maaløe et al., 2019). We use the baseline hierarchical VAE with 15 layers of latent variables, without the top-down and bottom-up connections. For the hierarchical VAE and

---

[2]Available at `https://cs.nyu.edu/~roweis/data.html`

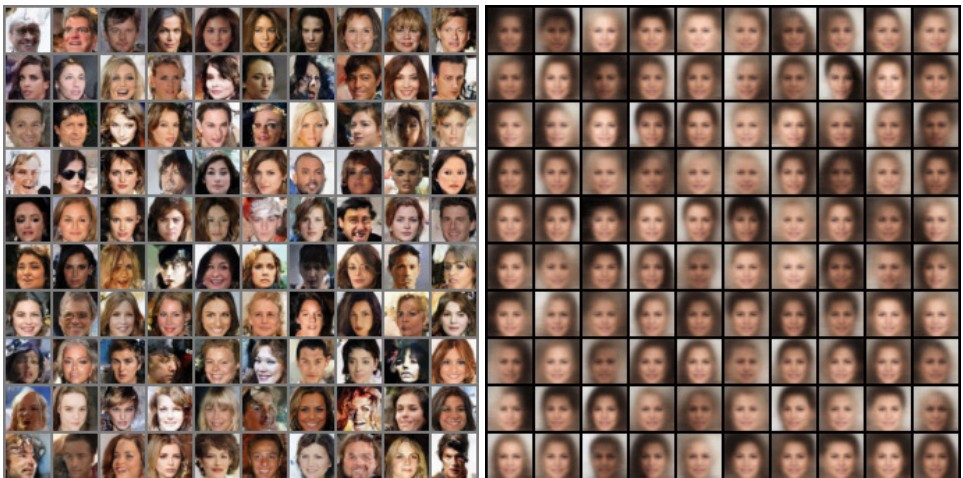

Figure 6: Samples from the $\sigma$-VAE (left) and the Gaussian VAE (right) on the CelebA dataset, images cropped to the face for clarity. The Gaussian VAE produces blurry results with indistinct face features, while the $\sigma$-VAE is able to produce accurate images of faces.

Table 3: Generative modeling performance of the proposed $\sigma$-VAE on CelebA, CIFAR, and Frey Face with a smaller model. We see that uncalibrated decoders such as mean-only Gaussian perform poorly. $\beta$-VAE allows to calibrate the decoder but needs careful hyperparameter tuning. Calibrated decoders such as categorical or $\sigma$-VAE perform best.

| | CelebA VAE | | CIFAR VAE | | Frey Face VAE | |
|---|---|---|---|---|---|---|
| | $-\log p \downarrow$ | FID $\downarrow$ | $-\log p \downarrow$ | FID $\downarrow$ | $-\log p \downarrow$ | FID $\downarrow$ |
| Bernoulli VAE Gregor et al. (2015) | | 102.7 | | 165.1 | | 47.7 |
| Categorical VAE | ¡10195 | **50.45** | ¡10673 | 124.1 | < **2454** | 50.16 |
| bitwise-categorical VAE | 11019 | 56.36 | 11604 | 99.65 | < 3173 | 66.77 |
| Logistic mixture VAE | ¡**10154** | 61.81 | ¡**10648** | 100.2 | < 2562 | 50.28 |
| Gaussian VAE | < 2201 | 144.8 | < 1409 | 205.8 | < 726.4 | 80.17 |
| $\beta$-VAE Higgins et al. (2017) | < −1942 | 58.73 | < −1318 | 117.9 | < −420.0 | **37.61** |
| Shared $\sigma$-VAE (Ours) | < −1939 | 73.27 | < −1830 | 137.8 | < −49.78 | 42.86 |
| Optimal $\sigma$-VAE (Ours) | < −1951 | 61.27 | < −1832 | **80.9** | < −1622 | 53.36 |
| Opt. per-image $\sigma$-VAE (Ours) | | **53.13** | | 89.88 | | 56.07 |

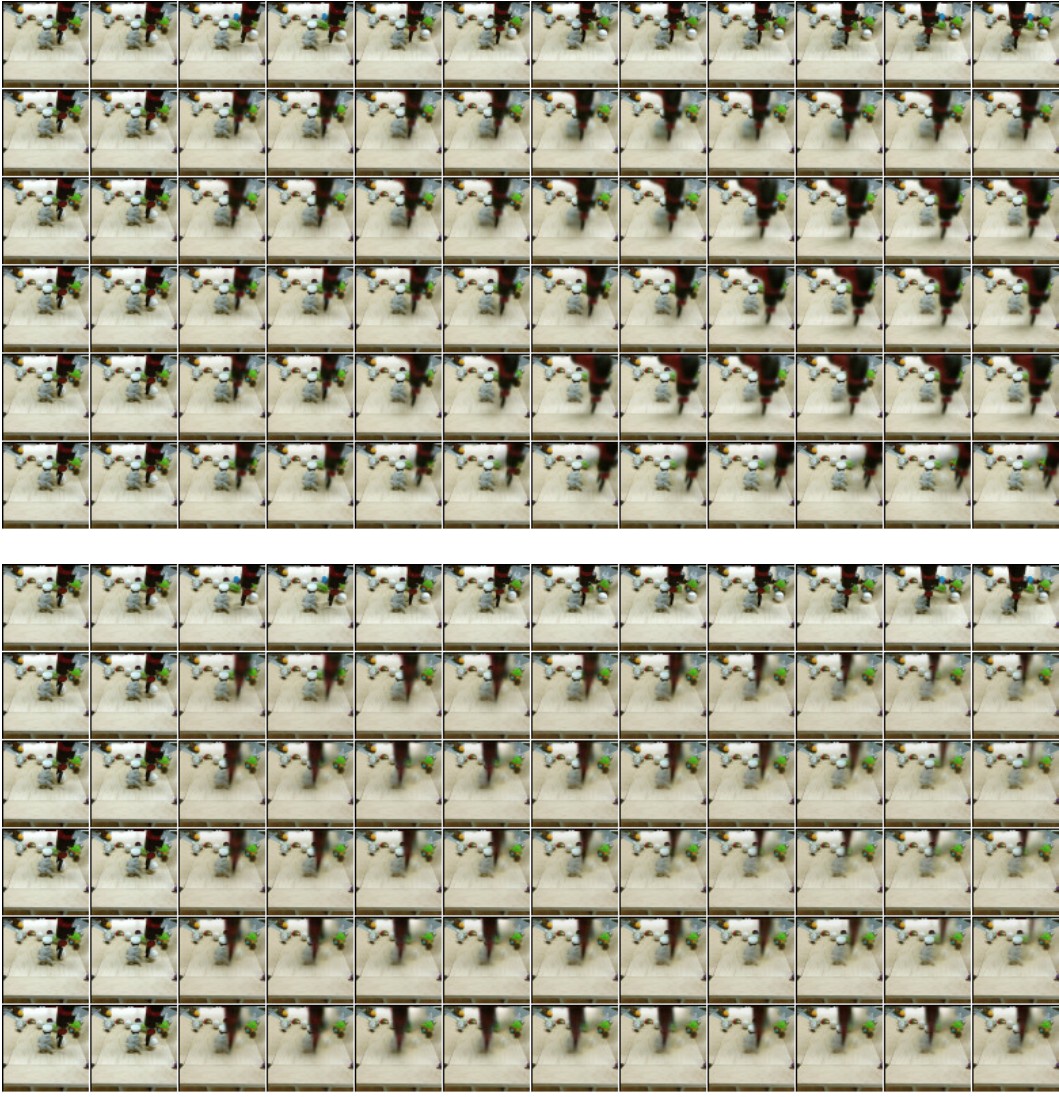

Figure 7: Samples from the $\sigma$-VAE (top) and the Gaussian VAE (bottom) on the BAIR dataset. Sampled sequences conditioned on two initial frames are shown, and the ground truth sequence is shown at the top. The Gaussian VAE produces blurry robot arm texture and the arm often disappears towards the end of the sequence, while the $\sigma$-VAE is able to produce sequences with realistic motion and model the details of the arm texture, such as the gripper.

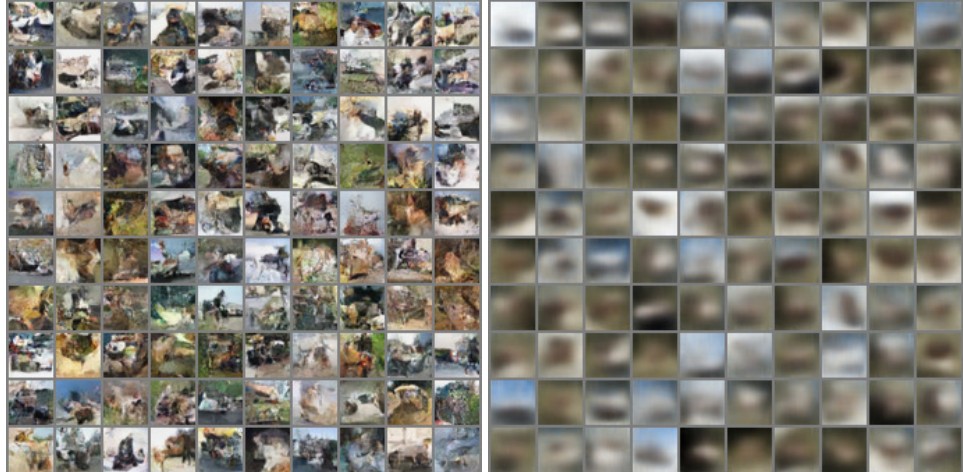

Figure 8: Samples from the $\sigma$-VAE (left) and the Gaussian VAE (right) on the challenging CIFAR dataset. The Gaussian VAE produces blurry results with muted colors, while the $\sigma$-VAE models the distribution of shapes in the CIFAR data more faithfully.

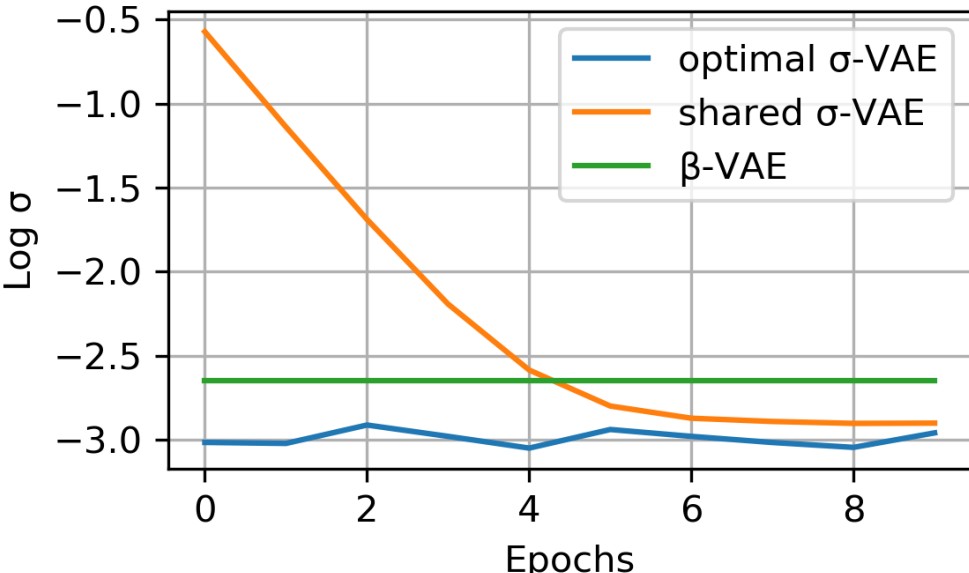

Figure 9: Variance convergence speed on SVHN. We see that the shared $\sigma$-VAE which optimizes the variance with gradient descent has an initial period of convergence when the variance converges to the region of the optimal value. In contrast, $\sigma$-VAE with analytical (optimal) variance quickly learns a good estimate of the variance, which leads to better performance. The unit variance Gaussian $\beta$-VAE can be interpreted as having a constant variance determined by $\beta$, shown here. Since the variance doesn't change throughout training, it achieves suboptimal performance.

the SVG-LP model, we use the default hyperparameters in the respective implementations. We use the standard train-val-test split for all datasets. All models were trained on a single high-end GPU. We use the official PyTorch implementation of the Inception network to compute FID. All methods are compared on the same hyperparameters.

## C  Empirical analyzis of approximations for optimal $\sigma$-VAE

The optimal $\sigma$-VAE requires computing the following estimate of the variance

$$\sigma^* = \arg\max_{\sigma} \mathbb{E}_{x \sim \text{Data}} \mathbb{E}_{q(z|x)} \left[ \ln p(x|\mu_\theta(z), \sigma^2 I) \right] = \mathbb{E}_{x \sim \text{Data}} \mathbb{E}_{q(z|x)} \text{MSE}(x, \mu_\theta(z)). \quad (7)$$

This requires computing two expectations, with respect to the data in the dataset, and with respect to the encoder distribution. We use MC sampling with one sample per data point to approximate both expectations. Inspired by common practices in VAEs, we use one sample per data point to approximate the inner expectation. On SVHN, the standard error of this approximation is $0.26\%$ of the value of sigma. We further approximate the outer expectation with a single batch instead of the entire dataset. On SVHN, the standard error of this approximation is $2\%$ of the value of sigma. We see that both approximations are accurate in practice. The second approximation yields a biased estimate of the evidence lower bound because the same batch is used to approximate the variance and compute the lower bound estimate. However, this bias can be corrected by using a different batch, or with a running average of the variance with an appropriate decay. This running average can also be used to reduce the variance of the estimate and to achieve convergence guarantees, but we did not find it necessary in our experiments.

## D  Alternative Decoder Choices

We describe the alternative decoders evaluated in Table 2: using the bitwise-categorical, and the logistic mixture distributions.

**Bitwise-categorical VAE** While the 256-way categorical decoder described in Section 3.2 is very powerful due to the ability to specify any possible intensity distribution, it suffers from high computational and memory requirements. Because 256 values need to be kept for each pixel and channel, simply keeping this distribution in memory for one 3-channel $1024 \times 1024$ image would require 3 GiB of memory, compared to 0.012 GiB for the Gaussian decoder. Therefore, training deep neural networks with this full categorical distribution is impractical for high-resolution images or videos. The bitwise-categorical VAE improves the memory complexity by defining the distribution over 256 values in a more compact way. Specifically, it defines a binary distribution over each bit in the pixel intensity value, requiring 8 values in total, one for each bit. This distribution can be thought of as a classifier that predicts the value of each bit in the image separately. In our implementation of the bitwise-categorical likelihood, we convert the image channels to binary format and use the standard binary cross-entropy loss (which reduces to binary log-likelihood since all bits in the image are deterministically either zero or one). While in our experiments the bitwise-categorical distribution did not outperform other choices, it often performs on par with our proposed method. We expect this distribution to be useful due to its generality as it is able to represent values stored in any digital format by converting them into binary.

Table 4: ELBO on discretized data. All distributions except categorical have scalar scale parameters. The $\sigma$-VAE performs well on the discretized ELBO metric, performing similarly to a discrete distribution parametrized as a discretized Gaussian or discretized Logistic. Full categorical distribution attains highest likelihood due to having the most statistical power.

| | CIFAR VAE | | |
| --- | --- | --- | --- |
| | $-\log \text{pdf} \downarrow$ | $-\log p \downarrow$ | FID $\downarrow$ |
| Categorical VAE | | $< \mathbf{10673}$ | **137.6** |
| Gaussian VAE | $< 740.5$ | $< 15131$ | 212.7 |
| Gaussian $\sigma$-VAE | $< -896.1$ | $< 11120$ | **136.7** |
| Disc. Gaussian $\sigma$-VAE | | $< 11117$ | **136.9** |
| Disc. Logistic $\sigma$-VAE | | $< 11103$ | **136.7** |

**Logistic mixture VAE** For this decoder, we adapt the discretized logistic mixture from Salimans et al. (2017). To define a discrete 256-way distribution, it divides the corresponding continuous distribution into 256 bins, where the probability mass is defined as the integral of the PDF over the corresponding bin. (Kingma et al., 2016) uses the logistic distribution discretized in this manner for the decoder. Salimans et al. (2017) suggests to make all bins except the first and the last be of equal size, whereas the first and the last bin include, respectively, the intervals $(-\infty, 0]$ and $[1, \infty)$. Salimans et al. (2017) further suggests using a mixture of discretized logistics for improved capacity. Our implementation largely follows the one in Salimans et al. (2017), however, we note that the original implementation is not suitable for learning latent variable models, as it generates the channels autoregressively. This will cause the latent variable to lose color information since it can be represented by the autoregressive decoder. We therefore adapt the mixture of discretized logistics to the pure latent variable setup by removing the mean-adjusting coefficients from (Salimans et al., 2017). In our experiments, the logistic mixture outperformed other discrete distributions.

