# OpenReview forum: "Simple and Effective VAE Training with Calibrated Decoders"
_ICLR.cc/2021/Conference — Reject_

### Official Review · AnonReviewer1 · 2020-10-27
**a simple but interesting discovery (not sure if there is similar work before this)**

**Rating:** 6
**Confidence:** 4

**Review:**

This paper discusses a well-known problem of VAE training that decoder produces blurry reconstruction with constant variance. While much existing work addressed this problem by introducing independent variance training (as of the original VAE model) or additional hyper-parameters, those approaches usually come with additional training/tuning difficulty and even break the ELBO assumption. This paper proposed a simple $\sigma$-VAE that addresses the above problem by optimizing a single variance variable. This also could be easily connected to the well known $\beta$-VAE works. The experiment results in Tables 2 and 3 show the proposed model obtains a better FID score than the existing works on multiple datasets.

In general, I am not surprised that shared variance would work well in practice as it is somewhat obvious as lots of works with beta-VAE to get better performance. However, it is surprising that no one aligns beta-VAE with the variance of observation before.

The most valuable knowledge I learned from this paper is written in one small section about variance implementation details where a new variable $lambda$ is introduced to avoid numeric problems. It is more helpful than the main claim of the paper somewhat in practice.

Pros
===
1. The paper is well written, and the authors provide sufficient evidence to show the proposed model works well in practice. The experiment design is also intuitive.
2. Mutual information analysis is great for reading.

Cons
===
1. The proposed approach is too simple to validate its novelty since it seems a rewritten of the existing formula by assigning the beta hyper-parameter with another meaning.
2. Figure 8 is hard to believe as Gaussian VAE on CIFAR-10 should perform much better than it is shown here. Many GitHub implementation shows far better-generated images.

Again, it is quite surprising that the method is not introduced before the year 2020. I have an impression where some of the VAE implementations did include things like proposed in this paper (but fail to find them now).

---

> ### Author Response · Authors · 2020-11-14
> **Author Response: Clarification of novelty and experimental setup**
>
> We thank the reviewer for the valuable and helpful feedback. The review raised several concerns about novelty and the experimental setup, to which we respond individually below. Please let us know whether we have sufficiently addressed the concerns, or whether any other concerns still remain.
>
> _“Many GitHub implementation shows far better-generated images” than the baselines._
> These implementations likely use $\beta$-VAEs, which indeed often produces comparable results to our method. However, in contrast to our method, the $\beta$-VAE requires extensive tuning of an additional hyperparameter, and does not correspond to a valid evidence lower bound. The baseline in Fig 8 uses a naive unit variance Gaussian decoder, which indeed performs extremely poorly.
>
> _“The most valuable knowledge I learned from this paper … $\lambda$ is introduced to avoid numeric problems”_
> We are happy if our paper turned out useful for the reviewer! Indeed, the main motivation behind the paper is to compile advice on simple and effective VAE training for practitioners. When training VAEs, we further encourage the reviewer to try learning a shared variance of the decoder, which removes the need for additional hyperparameter tuning in $\beta$-VAE.
>
> _“It is quite surprising that the method is not introduced before.”_
> We would like to clarify the novelty of our submission. As we note in the paper, some prior work used learning the variance of the decoder. Our contribution is to (i) analyze the different methods for learning the variance and present practical advice, and (ii) present a novel algorithm for optimizing a shared variance analytically, which indeed hasn’t been done in prior work despite the simplicity of this approach. We believe that these contributions will be helpful for practitioners exactly because of their simplicity and good performance.

---

> > ### Comment · AnonReviewer1 · 2020-11-23
> > **Thanks for the response**
> >
> > Thank you for your response to my questions. It addressed some of my concerns.

---

### Official Review · AnonReviewer4 · 2020-10-29
**Some useful ideas, but lacks solid experimental verification**

**Rating:** 5
**Confidence:** 4

**Review:**

This paper claims two contributions: 1. Proposes a connection between beta-VAE with fixed variance Gaussian decoder and VAE with variable variance Gaussian decoder 2. Proposes to optimize the variance of a Gaussian decoder

This paper uses a simple but useful method, but I have some concerns about novelty and soundness of the experiments. I will give a score on the low side for now.

Pro:

The connection between beta-VAE and variable variance Gaussian decoder is somewhat interesting.

The proposal that the variance can be adjusted for each individual image seems novel, but currently there does not seem to be a good theoretical/empirical justification for this.

The writing is quite good. Easy to follow and clear.

Con:

I think the main claimed contribution has limited novelty. The main difference between the current work and the prior common practice is to learn a single variance shared by all the pixels (instead of learning a variance per pixel). This by itself is fine, the lack of novelty can be made up by very convincing experiments and good practical guidance. My major concern is that the paper does not sufficiently justify its choice with experiments:

The baselines are surprisingly bad. For example, the paper says it uses the model in Maaloe et al, 2019 for celebA; comparing the samples in the current paper and the original paper, there is a very big difference. Granted, the original paper has a somewhat different setup, but the sample quality is too unreasonable to be convincing (I think VAE samples looked like that in 2013). In addition, the FID score of baselines also seem to be much worse than typically reported in the literature. For example, a FID score of around 50 is already typical for VAE models in 2019 (e.g. see Dai et al, 2019), while the current paper report an FID of 186 for baselines. Such a big discrepancy does not instill confidence that the baselines are properly trained with current recommended practices.

Following up on the above point, typically some kind of annealing of the beta parameter is the standard practice to avoid converging to a poor local minimum (which seems to be what is happening to the baselines here). It would be nice to have some discussion / comparison. In fact, I think it is completely fine if the argument is to show that reasoning effort to choose an annealing schedule do not lead to optimal results. However, I think this point does not come out from the experiments.

The criticized alternative (learning a variance per pixel) has been widely used in the literature and the samples look quite reasonable (e.g. see http://ruishu.io/2018/03/14/vae/) unlike the baselines in the paper. Granted there are a few implementation tricks (such as bounding the output to avoid numerical instability), but these are quite mild. It is difficult to conclude that the proposed approach uses less practical implementation techniques to achieve these results.

Minor comments:

The terminology calibration can have many different but precise meanings. For example, when people say a probability forecaster is calibrated, there is a specific property the forecaster must satisfy. I don’t think the usage of the terminology calibration matches any standard usage, and hence can be confusing.

The title is not very informative. In particular, with the current interpretation of "calibration" which is to "learn better probabilities", I think any VAE paper could have used that title.


---------

Thank you for the detailed reply and revisions. I have increased my review score because several of my concerns have been addressed. I am not entirely convinced that the baselines use current best practices, and several claims in the paper regarding the inductive biases and calibration. Nevertheless I think the algorithms proposed in the paper is practically useful.

---

> ### Author Response · Authors · 2020-11-14
> **Author Response: Additional experiment with KL annealing and clarification of experimental setup**
>
> Thank you for the detailed review and feedback. We believe that the main issues raised in the review pertain to our experimental evaluation. To address these concerns, we performed the suggested experiment with KL annealing, and further provide clarifications about our experimental setup below. Please let us know if this adequately addresses all of your reservations, or if there are other issues that remain to be addressed.
>
> _Annealing of the beta parameter is the standard practice_
> We ran the naive Gaussian baseline with annealing of the beta parameter from 0 to 1, both with a linear and an exponential schedule, but it does not seem to significantly improve performance of this baseline. The FID distance on the SVHN data for this baseline is 112.5 without scheduling, 109.7 with a linear schedule, and 108.9 with an exponential schedule, as compared to the much better score of 22.25 by sigma-VAE. We added a discussion of this to Sec 5. We believe this is because the issue is not simply a local minimum, but that the naive Gaussian VAE objective does not encourage learning informative representations.
>
> _Learning a variance per pixel … has been widely used in the literature_
> While the per-pixel Gaussian variance has been used in some early work on VAEs, we do not believe it is commonly used anymore. Indeed, the citation supplied by the reviewer states:
> “de facto practice when using Gaussian observation models is to set the decoder variance as a global hyperparameter.”
> That is, to use the naive Gaussian or the beta-VAE approach. We believe our results are in line with other work that used per-pixel Gaussian variance. We also want to point out that we do bound the variance for this baseline, as described in Sec 3.
> At the same time, many recent works, including the citation supplied by the reviewer, use the beta-VAE, which indeed often produces comparable results to our method. However, in contrast to our method, beta-VAE requires extensive tuning of an additional hyperparameter, and does not correspond to a valid evidence lower bound.
>
> _The baselines are surprisingly bad._
> We would like to clarify the experimental setup in our work. The main points of comparison are the commonly used naive Gausian VAE with fixed variance, and the $\beta$-VAE, which can be interpreted as tuning the variance. The naive Gaussian VAE indeed produces extremely poor results. $\beta$-VAE produces good results, sometimes on par with our method, but requires time-consuming manual tuning of the beta constant, while calibrated decoders do not. Maaloe’19 uses the logistic mixture decoder, which is also calibrated, and, as can be seen from Table 2, also performs well, although worse than our method. We stress that the goal of our work is not necessarily to establish SOTA on image generation, but to provide practical recommendations that hold for many scenarios, including e.g. both image generation and video prediction. We used the official implementations for both Maaloe’19 and Denton’18 baselines, and we believe our baselines are representative of the currently used methods.
>
> _a FID score of around 50 is already typical_
> We note that the FID score depends on several hyperparameters and is not necessarily comparable across different papers. In our case, we use a batch size of 50 to compute FID as we observed that the ranking of the methods does not depend on the batch size.
>
> _The proposal that the variance can be adjusted for each individual image seems novel, but currently there does not seem to be a good theoretical/empirical justification for this._
> We note that a theoretical justification for this model is very simple. Since the per-image sigma-VAE model is more expressive, it will always find a better (or at least the same) objective value, and will better fit the data distribution. Empirically, we observe that this model can perform better, although not dramatically. This is likely due to the fact that each image has roughly similar uncertainty. However, we expect this architecture to be useful in cases where uncertainty is different for each data sample, such as for conditional VAEs.
>
> _Interpretation of "calibration" which is to "learn better probabilities"_
> This was indeed our intention. In the usage of the term ‘calibration’ that we are aware of (https://en.wikipedia.org/wiki/Calibration_(statistics)#In_classification), it means producing accurate class probabilities. We generalized this term to the continuous setting, where we use it to mean ‘producing accurate probabilities of continuous variables’. In our paper, we analyze calibrated decoders, i.e. decoders that are expressive enough to represent the required distribution p(x|z).

---

> > ### Comment · AnonReviewer4 · 2020-11-23
> > **Thank you and the rebuttal addressed some of my concerns**
> >
> > Thank you for the detailed response. I think the rebuttal addressed some but not all of my concerns. In particular, what I still do not understand is the following:
> >
> > You say that per-image variance is better than using a single variance for all images, because it is more flexible. However, learning a pixel-wise variance is also more flexible than using a single variance for all pixels. I think there is a delicate situation of no-free-lunch and bias-variance trade-off here. I think the paper still lacks sufficient discussion of the inductive bias / bias-variance trade off of different parameterization choices for the variance term.
> >
> > Since you objective is sometimes identical to the beta-VAE objective (except your beta parameter is trainable), I don't see why beta-VAE doesn't correspond to a valid ELBO while your objective is a valid ELBO.
> >
> > I still don't agree with the usage of the term "calibration". For continuous distributions the term calibration has some very precise meanings (for example, see Gneiting et al, 2014). I don't think the usage here matches those either. It is already an incredibly overloaded terminology, and it's probably a bad idea to further confuse an already overloaded terminology. In geneal I think the description in title and introduction are a little oversold and should more accurately match the actual proposed contribution in the paper.

---

> > > ### Author Response · Authors · 2020-11-24
> > > **Author Response: added suggested discussion**
> > >
> > > Thank you for the additional useful and constructive feedback! We are glad that our initial response addressed some of the concerns. We have further updated the paper as follows.
> > >
> > > _I think the paper still lacks sufficient discussion of the inductive bias / bias-variance trade off of different parameterization choices for the variance term._
> > > Thank you for this insightful comment, we agree that this is a very interesting phenomenon which deserves further thought. We have added a discussion of the tradeoff to Sec 5.3.
> > >
> > > _I don't see why beta-VAE doesn't correspond to a valid ELBO while your objective is a valid ELBO._
> > > We would like to clarify our claim. The beta-VAE objective proposed in Higgins’17 is not derived from the ELBO, but instead is derived from constrained optimization with $\beta$ being the Lagrange multiplier. In terms of optimization, this objective is equivalent to sigma-VAE, however, it ignores the term $D \ln \sigma = D \ln \sqrt{\beta / 2}$. This term can be ignored if $\beta$ is a constant, but this leads to a wrong lower bound estimate. We have now clarified this in Sec 3. In practice, the error from ignoring this term often dominates the ELBO, rendering the estimate unusable. For instance, on SVHN, sigma-VAE achieves an ELBO of 3333, with $\sigma = 0.054$. Ignoring $D \ln \sigma$ yields the objective value of -3498 instead, a difference of 6831.
> > >
> > > _For continuous distributions the term calibration has some very precise meanings_
> > > We appreciate the provided reference and this insightful comment. While previously we used the term calibration without any clear definition, we agree that this is confusing. We have changed the Sec 3 heading, removed the vague statements from the submission, and added a discussion of the precise meaning in which we use the term to Sec 3 (we use it according to DeGroot’83, which is a similar definition to Gneiting’14). Specifically, in comparison to naive fixed Gaussian decoder, sigma-VAE is better calibrated as it learns a better uncertainty estimate (as opposed to simply better prediction). Learning this uncertainty is crucial for good performance as seen in Tab. 1, Fig. 3.
> > >
> > > Please let us know whether this resolves the remaining concerns. If there are any further issues with the paper, we will do our best to resolve them in the next revision.

---

### Official Review · AnonReviewer5 · 2020-11-05
**An analysis of learning variance in Gaussian decoders**

**Rating:** 6
**Confidence:** 5

**Review:**

**GENERAL**

The paper presents a more in-sight analysis of using learnable variance in Gaussian decoders in the Variational Auto-Encoder framework. The authors follow up on other papers in the literature and try to answer the following research question:
- If a Gaussian decoder is used, is it better to fix the variance (e.g., \sigma=1), or learn it?
In general, the paper is fine, however, it lacks novelty and it does not highlight that the VAE framework is a probabilistic framework, and a Gaussian decoder is not appropriate for modeling images. Nevertheless, I must admit that the analysis is properly performed.

**Strengths:**

S1: An in-depth analysis of using a shared variance across dimensions in Gaussian decoders.

S2: Comparing both NLL and FID is a good indication of showing importance of learning variance (or, more generally speaking, modeling uncertainty) in decoders in VAEs.


**Remarks:**

R1: VAEs constitute a sub-class of latent variables models with prescribed distributions (namely, all distributions are known in advance). In opposition to implicit models, the distributions must be picked accordingly to observed quantities (e.g., images). I am totally aware that many authors use Gaussian decoders for modeling images, however, doing it naively is simply wrong. The support of a normal distribution is [-\infty, +\infty], while images are typically represented by integers in [0, ..., 255]. Therefore, even if pixel values are normalized to [0, 1], they take one of 256 possible values. As a result, using Gaussian decoders is inappropriate.

It is possible to use dequantization as it is typically done in flow-based models, see:
- Theis, L., van den Oord, A., and Bethge, M. A note on the evaluation of generative models. ICLR 2016
- Ho, J., Chen, X., Srinivas, A., Duan, Y., and Abbeel, P. Flow++: Improving flow-based generative models with variational dequantization and architecture design. ICML 2019
- Winkler, C., Worrall, D., Hoogeboom, E., and Welling, M. Learning Likelihoods with Conditional Normalizing Flows. arXiv preprint, 2019
- Hoogeboom, E., Cohen, T. S., and Tomczak, J. M. Learning Discrete Distributions by Dequantization. arXiv preprint, 2020

However, this is not the case in this paper.
VAEs have a strong probabilistic foundation, and it should be treated as a probabilistic model. However, taking inappropriate distributions to model data, it results in:
(i) a wrong model;
(ii) propagating a wrong message in the deep learning community that any loss function works for VAEs.

I would highly appreciate if the authors would make it very clear in the paper, and start with a remark about choosing appropriate distribution for observed data. Afterwards, it could be explained that we need some sort of dequantization to utilize Gaussian decoders.

R2: The authors indicate a connection between a VAE with a Gaussian decoder and a \beta-VAE framework. The connection is very clear from the optimization perspective, but it is not the case from the modeling standpoint. If we take a look at the objective and consider the optimization process, then indeed, there is no difference in potential optima, because multiplying by \sigma or \beta results in the same objective. However, this has different consequences from the modeling perspective. \beta-VAE is a class of stochastic Auto-Endoders where the objective, the reconstruction error, is expanded by adding a Lagrangian multiplier. In other words, adding the equality constraint KL(q(z)||p(z)) = 0. However, the authors propose a valid VAE, i.e., the objective is a valid lower-bound to the log-likelihood function. I find it very confusing to indicate the connection without being very precise in what sense these two approaches are related to each other. Similarly to the remark R1, it could be very confusing in the DL community, and it could propagate a message that the VAE framework is a DL framework, while neural networks are an important (or even crucial) component of a probabilistic framework.

R3: As indicated by the authors, a similar approach was already discussed in the literature. Sharing a single variance across all dimensions was discussed in other papers. However, I must admit, not in-depth as here.

R4: It would be beneficial to discuss the following paper:
- Ghosh, P., Sajjadi, M. S., Vergari, A., Black, M., & Schölkopf, B. From variational to deterministic autoencoders. ICLR 2020

It is closely related and it could be see as an important other perspective on the problem.

*AFTER REBUTTAL*
I would like to thank the authors for their hard work! I increase my score to 6.

---

> ### Author Response · Authors · 2020-11-14
> **Author Response: Updated manuscript with the suggested discussion**
>
> We thank the reviewer for the insightful and useful suggestions, which we have incorporated in the manuscript as detailed below.
>
> _Gaussian decoders for modeling images_
> We thank the reviewer for this insightful comment and we have added a discussion on the choice of distribution to Sec 3.2, pointing out that discrete distributions are more suitable for discrete data. In practice we did not find dequantization necessary in our experiments, but we agree that the use of dequantization may improve results and provide better theoretical guarantees, and we added a discussion of dequantization to the related work. We note that in our work we are primarily concerned with the question of _how_ to learn decoders for continuous distributions. Dequantization is therefore largely orthogonal to our analysis, and we expect our approach to benefit from further improvements in dequantization.
>
> _Connection between a VAE with a Gaussian decoder and a \beta-VAE framework … valid lower-bound._
> Indeed, we believe that the main benefit of using calibrated decoders is the ability to leverage the correct probabilistic formulation, as we also state in the introduction. We have added this clarification to the \beta-VAE paragraph in Sec 3.1 as suggested.
>
> _A similar approach was already discussed in the literature_
> Prior work has indeed utilized many of the decoders we evaluate. As the reviewer points out, our paper goes in greater depth than prior work by analyzing these different choices and proposing a novel method for analytic variance optimization. We believe that our analysis of calibrated decoders is useful and timely as choosing a good decoder is crucial to the VAE performance.
>
> _Ghosh’19, “From variational to deterministic autoencoders”_
> We have added this relevant and interesting paper to the related work section. Ghosh’19 provides an alternative model that learns the prior distribution post-hoc, while we focus on the more conventional case of joint training.
>
> We would like to ask the reviewer to clarify whether we have adequately addressed the reviewer's concerns or whether there are any other concerns that prevent the reviewer from accepting the paper.

---

> > ### Comment · AnonReviewer5 · 2020-11-23
> > **After the rebuttal**
> >
> > Dear authors,
> >
> > Thank you for your detailed rebuttal. First of all, I highly appreciate your detailed answered and updates in the paper. I still believe that the novelty of the paper is rather limited, and I am still in doubt whether the paper propagates a proper statement about using Gaussian decoders for discrete data. However, as mentioned already, I really appreciate the hard work the authors put into answering and updating the paper. Therefore, I propose to raise my score to 6.
> >
> > Best.

---

> > > ### Author Response · Authors · 2020-11-24
> > > **Thanks for the update!**
> > >
> > > Dear reviewer, thank you for the useful feedback and the update! We believe the paper is much improved after the revision.

---

### Author Response · Authors · 2020-11-24
**General Author Response**

We thank all reviewers for the useful comments and suggestions. The reviewers noted the paper has “useful ideas”, and provides an “in-depth analysis” that is “great for reading”, but noted that the novelty of the proposed approach is limited. We have revised the paper to add the suggested discussion and improvements to experimental evaluation.

We further address the common concern about novelty here. Our contribution is to (i) analyze the different methods for learning the variance and present practical advice, and (ii) present a simple but novel algorithm for optimizing a shared variance analytically. We believe that our proposed method will be useful precisely because of its simplicity and good performance. We further believe that our analysis of this understudied problem is timely and necessary, given the importance of learning calibrated decoders for the performance of the VAE, and the recent trend to disregard calibrated decoders and use ad-hoc modifications to the objective instead. We believe that by providing this analysis, we can enable practitioners to make better choices of decoder distributions, improving results and reducing the need for hyperparameter tuning.

---

### Decision · Program_Chairs · 2021-01-07
**Final Decision**

**Decision:**

Reject

**Comment:**

This paper proposes to (re-)examine VAEs with calibrated uncertainties for the likelihood, which is say VAEs in which the variance is learned rather than chosen as a fixed hyperparameter. The authors argue that doing so provides a reasonable means of automatically navigating the tradeoff between minimizing the distortion (the reconstruction loss) and the rate (the KL loss) in the variational objective. In particular, the authors propose to use a diagonal covariance  Σ = σ^2 Ι that is shared across pixels, and note that it is trivial to define  σ(z) = MSE(x, μ(z)) on a per-image basis to minimize the reconstruction loss.

This is very much a borderline paper. Reviewers appreciate that the writing is clear, and acknowledge that revisiting the idea of learning calibrated is of interest to the community. At the same time, the reviewers note that the proposed approach has very limited technical novelty, and note problems with the experimental evaluation.

The metareviewer has read the paper, and is critical of the framing of this work. The manuscript in its current form does not do a sufficiently good job of discussing the large and detailed literature that exists on this topic. Learning calibrated decoders is by no means new, which this submission could and should acknowledge much more clearly. The two seminal papers on VAEs both considered learning calibrated decoders. Moreover there is a lack of thoughtful discussion of the reasons why learning a pixel-wise σ(z) is not common practice. The authors note that this can lead to problems with training stability, but fail to note that this problem is mathematically ill-posed; A well-known property of VAEs is that high-capacity models will memorize the training data, in the sense that the optimal learned marginal likelihood is equal to the empirical distribution over the training set (i.e. a mixture over delta peaks).

The metareviewer would expect to see a more thoughtful discussion of  the long line of work on navigating the trade-off between rate and distortion, as well as the role of model capacity. A good place to start would be a more careful discussion of the autoencoding and autodecoding limits (Alemi et al 2018) and the GECO paper (Rezende et al 2018). More broadly, the metareviewer would expect some discussion of approaches that improve the quality of generation such as [1], and work that considers effect of model capacity on generalization, such as [2].

In terms of experimental evaluation, this paper also somewhat falls short. As R4 notes, some of the results look worryingly bad, which may be due to the fact that the authors train for only 10 epochs (as indicated in  Appendix B). Moreover, what is once again lacking in experiments is a systematic consideration of the role of model capacity. Some comparison to more recent baselines than the β-VAE (e.g. GECO) would also be helpful here.

The metareviewer is sympathetic to the basic premise of this paper, which is the claim that learning a σ that is shared across pixels is a pretty good best practice in terms of finding a reasonable balance between rate and distortion. There is certainly room for a paper that communicates this idea. However, such a paper should (a) more explicitly position itself as revisiting this idea rather than introducing this idea, (b) include a more thoughtful discussion of related work, and (c) include a more robust empirical evaluation.

[1] Engel, J., Hoffman, M. & Roberts, A. Latent Constraints: Learning to Generate Conditionally from Unconditional Generative Models. arXiv:1711.05772 [cs, stat] (2017).

[2] Shu, R., Bui, H. H., Zhao, S., Kochenderfer, M. J. & Ermon, S. Amortized inference regularization. in Proceedings of the 32nd International Conference on Neural Information Processing Systems 4398–4407 (Curran Associates Inc., 2018).